# Gender Diversity in Academic Sector—Case Study

**Anna Wieczorek-Szymańska**

Faculty of Economics, Finance and Management, Institute of Management, University of Szczecin, 70-453 Szczecin, Poland; anna.wieczorek-szymanska@usz.edu.pl

**Abstract:** Diversity is one of the main characteristics of social groups, including work-teams. At the same time, gender is an important aspect of diversity in organizations, and gender diversity deals with the equal representation of men and women in the workplace. This article aims to analyze the issue of gender diversity in the academic sector and to evaluate the organizational maturity of particular universities in gender diversity management. To do so, the method of comparative case studies is used—Polish and Spanish higher education institutions are compared. First of all, the author describes the status of men and women in Poland and in Spain, in general (considering different socio-economic factors). In the next part of the article, the gender structure of employment in both the Polish and the Spanish academic sector is presented. Finally, the analysis of gender diversity in two universities is conducted. Additionally, the author introduces the model of organizational maturity in gender diversity management (OMDM), to evaluate organizational attitudes toward gender diversity and the type of gender diversity policy in universities. The findings reveal that, in both Polish and Spanish societies and economies, there still are barriers that cause inequalities between men and women in the labor market. Considering the situation in the academic sector, it can be said that the gender structure of employment is more balanced in Poland than in Spain. At the same time, the highest positions of full professors are mainly occupied by men both in Poland and in Spain. When analyzing the situation in the organizations, employment is more diverse in the Polish university, but both universities face the same problem—too little representation of women in top job positions. Consequently, both institutions are classified as those which are in the preliminary stage in the model of gender diversity management. This study contributes to a better understanding of the issue of gender diversity by comparing the status of men and women in the academic sector in two countries and in two universities. Additionally, the model of OMDM presented in this article can be a useful tool to assess the policy of gender diversity in different organizations.

**Keywords:** heterogeneity of work-teams; diversity management; gender; model of gender diversity management

## 1. Introduction

The issue of gender can be considered in a biological context (nature determines an individual's sex and, by this, defines a person as a man or a women) and in a social context (focusing mainly on defining social roles that are typical for a man or a woman in a society). Those roles are strengthened by the culture and the tradition of the society the person lives in. In turn, they affect attitudes and behaviors a person presents in his/her professional life. In many cases, it also leads to stereotypes and assigning people specific abilities, expectations and behavior only on the basis of the gender.

For many years, women were a minority in the labor market, since in most societies their main role was not to earn money but to run a household. They also did not participate in a political life, as, for example, women only received the right to vote in 1918 in Poland (in 1931 in Spain). However, times have changed. Women have fought for their right to choose whether to stay home or to work

and build their own career path. It was not, and still is not, easy to choose building a career, as women face serious problems in the labor market.

The author of the article has decided to take up the subject of gender diversity in modern organization. The research question is: What is the stage of gender diversity management in organization from the academic sector considering the model of organizational maturity in diversity management? Moreover, the main object of the research is to analyze the issue of gender diversity in the academic sector and to evaluate organizational maturity in gender diversity management. The objects of the research are Polish and Spanish universities. The research is a part of a wider project that aims at the comparison of gender structures in universities across Europe. The reason for choosing Polish and Spanish cases is that the population in both countries is on a similar level, as both countries have a similar gender structure and some similar problems of gender equality (like wage gap, lower estimated income and low percentage of Members of board).

The theoretical part of this paper is based on the literature review (the Web of Science Core Collection, EBESCO and SpringerLink databases were used as the main sources of data). The aim of the topic search (that the author carried out in the period of October–December 2019) was to identify publications for the phrases "gender diversity", "diversity management" and "diverse in the academic sector". The empirical part is based on the method of comparative case study (Goodrick 2014) and on the author's own model of organizational maturity in gender diversity management (Wieczorek-Szymańska 2017).

In this article, the author presents the justification of the conducted research and then the literature review. After that, the author presents the methods and materials used in the research. The next part describes the assumptions of the model of organizational maturity in diversity management and focuses on two case studies (Polish and Spanish universities). The article concludes with the summary.

The justification of the research conducted by the author is as follows:

1. Gender is one of the aspects of diversity in organizations, and gender diversity deals with the equal representation of men and women in the workplace. A work-team is a social group within which people should work collectively to achieve a synergic effect (Seroka-Stolka 2016). Understanding the factors of the diversity in a work environment is crucial to underline the source of people's behavior and to foresee its impact on collective work.

2. The issue is important, as there are barriers for women in the labor market. In the European Union, the phenomenon of women's presence on boards of directors has been present since 2010, when the European Commission introduced the "Strategy for equality between women and men 2010–2015". One of its targets was to have 40 percent of women represented in committees and expert groups established by the Commission (European Commission 2011). Moreover, in November 2012, the Commission proposed another target of 40 per cent representation of women in the board of directors in stock-listed companies in Europe by 2020. Despite the policy in Western Europe, only 17 percent of executive-committee members are women, and women comprise just 32 percent of members of corporate boards for companies listed in Western Europe's major market indexes (exhibit) (Devillard et al. 2016). Additionally, for example, women still work more unpaid hours than men and suffer from gender pay gaps (Arulampalam et al. 2007). It seems that stereotypes and prejudices remain as the main barriers in terms of gender diversity in organizations. Discrimination concerning gender, namely unequal treatment of women, is specifically evident, which is not justified by any objective reasons (Wieczorek-Szymańska and Leoński 2017). What is more, women are overrepresented in so called "feminized professions", like teachers and nurses (Kaźmierczyk and Żelichowska 2017), but face a "glass ceiling" in professions dominated by men (Furmańczyk and Kaźmierczyk 2017). That is why there is a need to join the discussion about gender diversity and its impact on the work environment.

There is still not enough scientific evidence of the gender diversity issue in the higher-education sector. Bradley et al. (2009) even state that there is not much focus on women's problems of

representation in the higher-education sector. In turn, Patterson et al. (2009) focused on numerous gaps, like salary, promotion, discrimination and harassment, that women experience in academic leadership. What is more, according to Cornwell and Kellough (1994) or Riccucci (2002), diversity management in the public sector is mainly dominated by Equal Employment Opportunity (EEO) and Affirmative Action (AA) approaches compared with business case approach in the private sector. The literature on the gender diversity management (GDM) is quite rich but focuses mainly on the issue of gender diversity and its relationship with organizational performance (Dwyer et al. 2003; Choi and Rainey 2010; Richard et al. 2013; Velte et al. 2014; Chin and Tat, 2015; Opstrup and Villadsen, 2015; Adusei et al. 2017; Cho et al. 2017). Other researchers discuss the problem of gender diversity in the context of a firm's innovation orientation (Richard et al. 2004; Talke et al. 2011) or analyze the issue in the context of productivity (van Knippenberg et al. 2011; Sabharwal, 2014; Parola et al. 2015; Ali, 2016). The issue of gender is also described in terms of the so-called "glass ceiling"—discrimination in the promotion of employees to the top-level positions in organizations (Akpinar-Sposito, 2013; Powell and Butterfield, 2015; Keenawinna and Sajeevanie, 2015; Roman, 2017).

It is interesting to note that only some research on diversity management (DM) has been done in reference to the academic sector. For example, Shackleton (2007) conducted a study to observe how gender discrimination influences promotion aspects in the academic sector. Research on gender segregation also shows how social norms and expectations encourage gendered career choices in higher education (Cech 2013; Charles and Bradley 2009). In addition, the "She Figures Report" (Directorate-General for Research and Innovation European Commission) describes the issue of gender equality in research and innovation (R&I) in Europe. This cross-national research looks for the differences in the experiences of women and men hired in the academic sector by exploring pay gap, working conditions, success in obtaining research funds, and the percentage difference of women to men as authors of scientific publications and as patent inventors. Another cross-national research study was done in 2019 on a group of 159 European higher-education institutions. The report presents key evidence about how universities could and do promote the idea of diversity, equality and inclusion (Claeys-Kulik et al. 2019).

Other publications focus on women's situation in science, technology, engineering and mathematics (STEM). For example, UNESCO's (United Nations Educational, Scientific and Cultural Organization) report aims to identify factors that hinder or facilitate women's achievements in STEM. It presents results from more than 120 countries and suggests what could be done to inspire, engage and empower women in STEM (UNESCO 2017). In turn, Howe-Walsh and Turnbull (2016) report on women's experiences regarding the perceived barriers to leadership positions in science and technology (ST) in UK universities. The author identifies the organizational barriers, such as temporary work arrangements, male-dominated networks, intimidation and harassment, as well as individual factors, such as lack of confidence. Botella et al. (2019) conducted research on gender diversity in STEM disciplines and identified some of the main problems that women face in their professional careers through different age stages, considering the Spanish case.

Vermeulen (2010) examines different aspects of DM in higher education system by comparing a South Africa Case Perspective to German Case Perspective. He presents both students' and personnel's perspective in DM. Meric et al. (2015) present innovative policies regarding the DM in a higher education institution on the example of United States Air Force Academy (US AFA). In Polish literature, Nowakowska-Grunt and Kabus (2014) discuss DM in the academic sector by analyzing the case study of Technical University in Czestochowa, and in Spain, Vázquez-Cupeiro and Elston (2006) analyzed the academic career trajectories in reference to gender.

Following Bradley et al. (2009), it is interesting to note that there is not much focus on the women's problem of representation in the academic sector. Hence, there still is a gap in scientific research on the issue and the need to start the discussion on women's situation in universities. What is more, there is no research on organizational maturity in DM. At the same time, we must remember that the academic sector faces new challenges and changes now and into the future (Achim et al. 2009).

## 2. Material and Methods

The question for the purpose of the research is as follows: What is the stage of gender diversity management in the organization, considering the model of organizational maturity in diversity management? The main object of the research is to analyze the issue of gender diversity in the academic sector and to evaluate organizational maturity in gender diversity management. In order to achieve the aim of the paper, the following operational objectives have been set: to discuss the idea of gender diversity management in an organization; to present the model of organizational maturity in managing diversity; to overview the socio-economic situation of women in both countries; and to analyze the gender structure of employment in the universities and action undertaken in gender diversity management in the organizations.

The author uses a comparative case study method for the purpose of the research. The qualitative method is used, as the author's aim was to understand the nature of the phenomenon (GDM in modern organizations). The researcher collected the data from the secondary data analysis and primary data analysis. Secondary data were gathered mainly through related research articles, reports, organizational documents and websites. Primary data required for the study were gathered from a questionnaire. The survey questionnaire was composed both in Polish and English and consisted of two sections, A and B. Section A refereed to the structure of employment in the university. Section B contained three questions related to GDM in organization.

The first question verified how gender diversity among teachers in the university is reached. The possible answers were as follows: (1) The diversity is a result of compliance with the law, (2) the diversity is a result of strong organizational commitment to the idea of diversity, (3) the diversity is a result of demographic situation in the labor market and (4) the diversity is not important at all for the organization.

The second question examined what kind of policy was introduced in the university to secure gender diversity. The respondent could choose one of four responses: (1) There is no specific gender policy in the university, (2) the policy is a result of antidiscrimination law, (3) the special antidiscrimination internal regulations are created and introduced in the university or (4) the policy is based on quotas.

The third question analyzed the organizational attitude toward the issue of gender diversity. Each respondent could choose one of the following answers: (1) Gender diversity is a source of interpersonal differences that cause higher risk of conflict; (2) gender diversity is an effect of the diverse labor force; (3) gender diversity brings only benefits like a better organizational image, more creative work-teams and better efficiency; or (4) gender diversity is a source of both opportunities and threats for organization, so it needs an active management. The survey was conducted in the universities in 2019–2020. In both cases, respondents were top managers from the human resources department of the universities. In the case of the University of Szczecin, the respondent was Vice-Rector; in the case of the University of Cordoba, the respondent was the head of the HR department. Apart from the questionnaire, the researcher also conducted participative observation (the author has been working for the University of Szczecin since 2007 and was a participant of the Erasmus plus Program in the University of Cordoba in 2019. During the ERASMUS scholarship, the author had a chance to observe work organization processes in the University of Cordoba).

The reason for choosing the universities as the subjects of this research is that the author of the article had a chance to work at both of them, and so had an opportunity to do some research by active participation. What is more, the universities are in some respects similar (similar size, similar date of establishment and similar number of faculties), and in some other ways, they are different (different countries, the number of people in both cities is different and the socio-economic conditions are different). For the reasons mentioned, both organizations are ideal subjects for the comparative case study.

### 3. Analyzing Organizational Maturity in Gender Diversity Management

*3.1. Conceptual Model of Organizational Maturity in Gender Diversity Management*

The concept of diversity management has its roots in Equal Employment Opportunity (EEO) and Affirmative Action (AA) approaches. In turn, these concepts were the outcomes of the Civil Rights Movement that took place in the USA, in the 20th century (Shore et al. 2009).

According to Crosby and Konrad (2002), AA occurs when an organization introduces antidiscrimination programs (that prevent evaluating employees on the basis of personal characteristics, such as gender, race and age, and instead on their ability to perform their job) and the equal opportunity exists (equal access to promotion, jobs and salary for people of different sex, race and age). In the case of AA, the target groups are, for example, ethnic, race minorities or women (Besler and Sezerel 2012). Managing diversity includes EEO and AA approaches, but the scope is a lot broader. While EEO and AA are the product of the Civil Rights Movement of the 1960s and legislation, DM could be named as the second generation of EEO (Shen et al. 2009).

Modern DM was developed in 1980 as a reaction to the demographic changes of the USA labor market and consumer market. In 1987, the report "Workforce 2000", by Hudson Institute, presented demographic shifts according to which white males would no longer be a majority in the workforce due to the total increase of women and other minorities in the labor market (Johnston and Packer 1987). In consequence, research on diversity began to focus on work-teams or the business case for managing the workforce. Since the 1990s, DM has become more popular in Europe and has emerged as a key issue in international business.

The basic assumption in the concept of DM is that work-teams consist of a diverse population. There are different aspects of the diversity, like age, sex, ethnicity, race and disability, but also the personality or the style of work (Lawthom 2003). Some factors can be shaped by individuals, while some others are beyond individual's control. Some of them affect how a person is perceived by a social environment, while some others can be a source of discrimination in the workplace. For this reason, Cox (1991) assumes that DM is a way of human resources management which aims to maximize the potential advantages of diversity and minimize its disadvantages. In this context, it is suggested that workforce diversity may provide organizations with a valuable, rare, inimitable and non-substitutable competitive advantage (McMahan et al. 1998). According to Thomas (1990), managing diversity in an organization involves managing people's differences, in order to enable all groups of employees to succeed in performing the job, so the goal is to create a modern working environment where everyone feels valuable in their jobs.

The importance of DM arises, as it has a strong impact on organizational outcomes, both on the individual and organizational level (Foldy 2004; Pits and Jarry 2007). For the purpose of the paper, it is assumed that DM is about creating such conditions within the organization that allow people to build and use unique competences thanks to a diverse workforce. Hence, there is a demand for active DM.

Depending on the type of actions carried out in an organization, one can consider a different orientation of the organization toward DM (Table 1). Podsiadlowski et al. (2009) define five basic perspectives that can be introduced within the DM in an organization. These are reinforcing homogeneity, color-blind approach, fairness, the access, and integration and learning perspective. The approaches mentioned above present managers' attitudes toward DM, starting from defensive perspective, which is reinforcing homogeneity (one end of the continuum), to a proactive perspective, integration and learning (the other end of the continuum). Moore (1999) presents a similar point of view as he mentions different managers' orientation to workforce diversity: The antagonistic approach is typical for the strategy of creating a so-called monolithic organization which does not comply with a comparatively diverse workforce in reality (Cox 1991), while a realistic one characterizes managers acting in accordance with modern DM approach. Konrad et al. (2016) discuss the idea of DEMS—diversity and equality management system—which is a set of diversity and equality management practices aligned with the business strategy. The process of DM policy can also be

described by three other models, namely adaptive, consolidation and business (Jamka 2011; Urbaniak 2014). The adaptive model mainly follows the rules that are typical for EEO; the consolidation model treats DM as an element of Corporate Social Responsibility action, while the business model is compatible with modern DM.

**Table 1.** Approaches to diversity management (DM).

| Author Name/s | The Name of the Model and Main Characteristics |
| --- | --- |
| Podsiadlowski et al. (2009) | **Reinforcing Homogeneity:** avoiding or rejecting diversity in favor of homogeneous workforce.<br>**Color-blind approach:** practices preventing discrimination. The approach is typical for equal employment opportunities.<br>**Fairness:** ensuring equal and fair treatment by supporting minority groups or reducing social inequalities.<br>**The access perspective:** underlines the ability to access diverse customers by hiring people who reflect the customer market.<br>**Integration and learning perspective:** Both the organization, as a whole, and employees benefit from diversity, as it creates a learning environment where mutual acceptance of minority and majority groups alike exist. |
| Moore (1999) | **Antagonistic approach:** It emphasizes differences and problems resulting from diverse groups of employees, e.g., the risk of conflicts and lack of common organizational identity.<br>**Neutral acceptance of diversity:** Diversity in the organization is assumed to be natural, as a result of demographic changes. No particular benefits of having diverse staff is perceived, so no specific management procedures are conducted.<br>**Naive positive attitude:** Managers affirm differences between people and expect automatic benefits from hiring gender-diversified employees. In consequence, they ignore the challenges of gender diversity management.<br>**Realistic approach:** emphasizes the need for active management of gender diversity in the organization, in order to achieve the intended results. Both the benefits and the possible problems resulting from hiring gender diversified employees are taken into account. |
| Konrad et al. (2016) | **Classical disparity structures:** Organizations are too small to be covered by employment equity legislation or are not federal contractors, so without institutional pressure, decision makers can deny the existence of labor-market discrimination. As a result, DEMS include relatively few employment equality, diversity or inclusion practices.<br>**Institutional DEMS:** includes multiple activities to remove barriers to diversity hiring.<br>**Configurational DEMS:** covers multiple aspects of the business case for diversity, as well as institutional requirements. |
| Jamka (2011); Urbaniak (2014) | **Adaptive model:** removing barriers to access, and observance of antidiscrimination law.<br>**Consolidation model:** Diversity is a part of the strategy of Corporate Social Responsibility, so the organizational culture is diversity oriented. The idea of diversity is used to access diverse customers.<br>**Business model:** Diversity is perceived as a source of competitive advantage for an organization. The analysis of effectiveness and the balance of costs/benefits of DM programs are conducted. |

Notes: DEMS = diversity and equality management systems. DM = diversity management.

Although DM deals with different aspects of the diversity in a workplace, the article focuses on one particular dimension—gender diversity.

Gender equality and women's right are one of the fundamental principles of the European Social Charter, which was established in 1961, by the European Commission (European Social Charter). It includes many recommendations on the equal treatment of men and women in social, political and business life. For example, the recommendation on gender mainstreaming (Recommendation No. R 98-14) calls on member states to create an enabling environment to facilitate conditions for the implementation of gender streamlining in public sector; the balanced participation of woman and men in political and public decision making (Recommendation Rec 2003-3) calls on the balanced participation of men and women in decision-making bodies in political and public life; gender equality

standards and mechanisms (Recommendation REC 2007-17) provide an extensive list of measures to achieve gender equality in practice. All those legal acts prove the importance of equal opportunity policy in Europe. The question arises how organizations, both private and public ones, introduce the issue of gender into the management process.

The organizational attitudes toward gender diversity range from intolerance to tolerance (Joplin and Daus 1997). Therefore, the author of the article proposes to create the model of OMDM which reflects top management's commitment to diversity management and the type of the policy that promotes the diversity in the organization.

The assumptions of the model are as follows:

1.  The model is a sort of continuum on which one can place an organization according to two factors (Figure 1): organizational attitudes toward GD of employees, which vary from neutral to proactive, and the type of GD policy which varies from compliance with antidiscrimination law to strategic policy.
2.  Each organization that meets at least two conditions, namely neutral attitude toward diversity and the observance of antidiscrimination law, can be place on the continuum. The location on the graph depends on the type of activities undertaken within DM.
3.  The organization that does not fulfill the conditions mentioned in the second point cannot be taken into consideration in the model of maturity in DM, as they do not manage diversity at all.

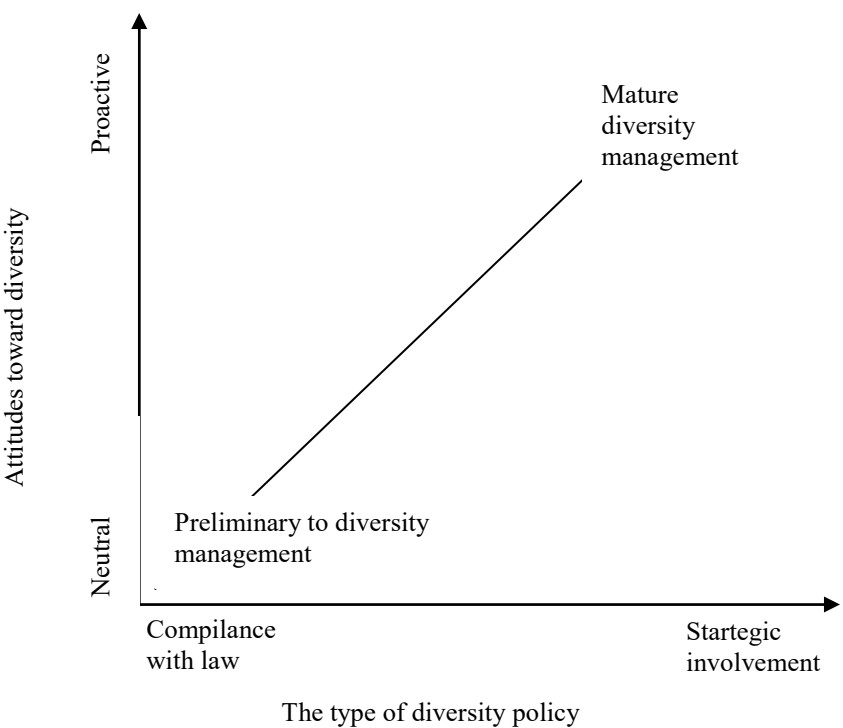

**Figure 1.** The model of organizational maturity in managing diversity.

The first stage of gender diversity management in an organization is so-called preliminary to DM. The institutions that reach the preliminary phase present neutral attitudes toward diversity issue and operate accordingly to law regulations. This means that the structure of employment is an effect of demographic changes both in labor and customers' markets. In managers' opinion, diversity is neither good nor bad, so no specific action occurs to hire or manage diverse employees within an organization. At the same time, the organization does not break the antidiscrimination law.

In turn, the mature GDM is characterized by a proactive attitude toward workforce diversity and strategic diverse policy. Mature GDM means a strong commitment to diversity, as diversity is a part

of organizational culture, not just a program of management. All employees are treated as diverse. The idea is that everybody learns from each other. Diversity is a source of competitive advantage and tied directly to organizational vision, mission and strategy. Visible and active management involvement is crucial, as well as setting clear targets of DM and evaluating the effectiveness of the plans. Team-building and group process training are emphasized within company. Organizations reach diverse consumer markets and have diverse group of suppliers (Slater et al. 2008). Mature GDM is typical for the so-called diversity-oriented organization, which, according to Kandola and Fullerton (1998), is characterized by six elements. First of all, it has a strong and positive mission, where diversity is one of the long-term goals. Second of all, organizational objectives are formulated to promote fairness and to avoid the domination of a particular group of employees at any organizational level. Third of all, the organization hires skilled people who are aware of the effects of biases in the process of decision-making. The way the work is organized is flexible. All individuals have access to promotions and self-development. Finally, the organizational culture is based on trust and the absence of discrimination. To reorient an organization from preliminary stage toward the mature GDM, managers must be aware that compliance with antidiscrimination law is necessary but not a sufficient condition. The more proactive attitude of organizational managers toward diversity, and the stronger strategic importance of diverse workforce, the closer to mature-oriented DM the organization will be. The idea of the continuum in DM approach helps to place an organization closer either to a preliminary stage or to a maturity stage in DM. Of course, the limitation of the model is that it presents too simplistic a view; for this reason, it should be verified in practice, to check its usability.

*3.2. Research Findings*

3.2.1. Case Study 1: The University of Szczecin

Socio-Economic Conditions in Poland

According to the World Economic Forum, Poland was ranked 38 worldwide in gender equality[1] for 2016, but dropped to 39 in 2017 and to 42 in 2018 (Worldwide Economic Forum 2016, 2017 and 2018). The population sex ratio (female/male) shows that there are more women than men in Poland, and 62.5% of women who are of working age (15–64) engage actively in the labor market (in comparison, the ratio for men is 76.1%). One of the biggest problems of inequality between men and women in Poland is a wage gap for similar work, as the subindex is 0.56[2] (which shows the existence of a large disproportion in wages between men and women in Poland). Additionally, the estimated earned income (US$) for women is about 35.16% lower than for men. What is more, monthly earnings (means earnings of employees in local currency in nominal terms) is also lower for women in comparison to men, by about 17.78%. The ratio of women to men employed in senior roles[3] is 0.7 (41.2% of all senior positions are occupied by women), which again shows the domination of men. At the same time, women work more than men by about 32.4 minutes per day and spend more time in unpaid work—the average minutes of unpaid work for women is 60.0, and for men, it is 34.1 per day. In reference to the issue of economic leadership, more men (80%) than women (20%) are among the members of the board of directors of each company in the OECD ORBIS database and only in 26% of firms in private

---

[1]   The Worldwide Economic Forum publishes a yearly Global Gender Gap Report ranking 149 countries in the world, in order of how equally women are treated compared to men, with respect to economic participation and opportunities, educational attainment, health and survival, and political empowerment. The country classified as the first is the most equal in treating women to men.

[2]   The subindex is based on the survey and normalized on a 0–1 scale, where 0 means that, for similar work wages, men and women are not at all equal; 1 means that, for similar work wages, men and women are fully equal.

[3]   Senior roles are employees who plan, direct, coordinate and evaluate the overall activities of enterprises, governments and other organizations, or of organizational units, formulate and review their policies, laws, rules and regulations.

sector females are top managers. On the other hand, in 68% of firms in the private sector, women are among the principal owners (Worldwide Economic Forum 2018).

The Characteristics of the Polish Academic Sector

In Poland, most women graduate in business, administration and law (24.6% of total number of graduates are women, and 21.8% are men), while most men graduate in engineering, manufacturing and construction (9.4% of total number of graduates are women, and 26.1% are men). Women are interested the least in studying information and communication technologies (0.9% of women and 7.2% of men). At the same time, women are better educated in Poland, as 26% of them have tertiary education (ISCED 5-8)[4].

At the end of December 2016, 95,400 academic teachers were employed (95.8% as full-time employees), 45% of whom were women in the academic sector. The structure of full-time employed academic teachers considering job positions in the sector is presented in Table 2 (Central Statistical Office 2017). The employment in higher schools is gender balanced in the case of tutors, assistant lecturers, senior lectures and instructors[5]. On the other hand, in the case of the positions of professors and assistant professors, about 70% of employees are men. In contrast, most lecturers, lectors and librarians (marked as "other: in the table) are women (Table 2).

**Table 2.** The structure of employment, considering job positions in the higher education sector, in the academic year 2016/2017, in Poland.

| | Professors | Assistant Professors | Tutors | Assistant Lecturers | Senior Lecturers | Lecturers | Lectors | Instructors | Other |
|---|---|---|---|---|---|---|---|---|---|
| Grand total | 22,844 | 519 | 39,332 | 11,060 | 10,670 | 4900 | 1009 | 724 | 322 |
| Of whom are women (%) | **27.9** | **31.0** | **47.5** | **53.7** | **50.9** | **60.3** | **79.5** | **56.9** | **82.6** |
| Public higher schools | 18716 | 392 | 34,842 | 10,166 | 9992 | 4041 | 879 | 672 | 272 |
| Of whom are women (%) | **28.8** | **32.4** | **47.8** | **53.5** | **51.1** | **61.7** | **80.2** | **56.8** | **82.4** |
| Non-public higher schools | 4128 | 127 | 4490 | 894 | 678 | 859 | 130 | 52 | 50 |
| Of whom are women (%) | **23.8** | **26.8** | **45.3** | **55.8** | **47.4** | **53.9** | **74.6** | **57.7** | **84.0** |

Source: based on Central Statistical Office, 2017.

Gender Diversity Management in the University of Szczecin

The University of Szczecin (US) is a public university in Western Poland. It is the biggest university in the region of West Pomerania. It was established in 1984 and since then has been a very important entity in the socio-economic environment for the region. The University is a member of the European University Association (EUA) and holds full rights of an autonomous university. The educational offer is wide and includes 100 courses of study within 11 faculties, namely Philology (FP), Humanities (FH), Physical Education and Health Promotion (FEHP), Economics and Management (FEM), Mathematics and Physics (FMP), Geosciences (FG), Biology (FB), Law and Administration (FLA), Theology (FT), Economics of Services (FES), Socio-Economic in Gorzów Wielkopolski (FSEG) and Inter-Faculty

---

[4]   ISCED is International Standards Classification of Education. 5-8 means Short-cycle tertiary education, Bachelor or equivalent, Master or equivalent, Doctoral or equivalent.

[5]   The career-path in academic sector in Poland is as follows: 1. Assistant Lecturer (Asystent)—an early stage of an academic career after completing the Master Desgree or shortly after PhD 2. Tutors (Adiunkt)—PhD is mandatory, 3. Associate Professor (Profesor nadzwyczajny)—the stage after the habilitation, 4. Full Professor (Profesor zwyczajny)—the highest position full capacity for learning and research. There are some other positions like: Senior Lecturer (Starszy wykładowca) and Lecturer (Wykładowca)—these are positions for employees with or without PhD and the main responsibility of the employees is didactic.

Department (ID). The academic staff carry out different types of research in humanities, social, natural, technical, exact, medical and health sciences, as well as physical culture (About the University).

At the end of May 2018, 1097 academic teachers were employed at the US (98.36% as full-time employees, and 1.64% as part-time employees). In total, 597 of all employed were women (54.42%), and 500 of them were men (45.58%). Thus, it can be said that, in general, the gender structure of the employment is balanced. On the other hand, the analysis of the employment structure in individual faculties in the US shows that not all of the units are gender diverse (Figure 2).

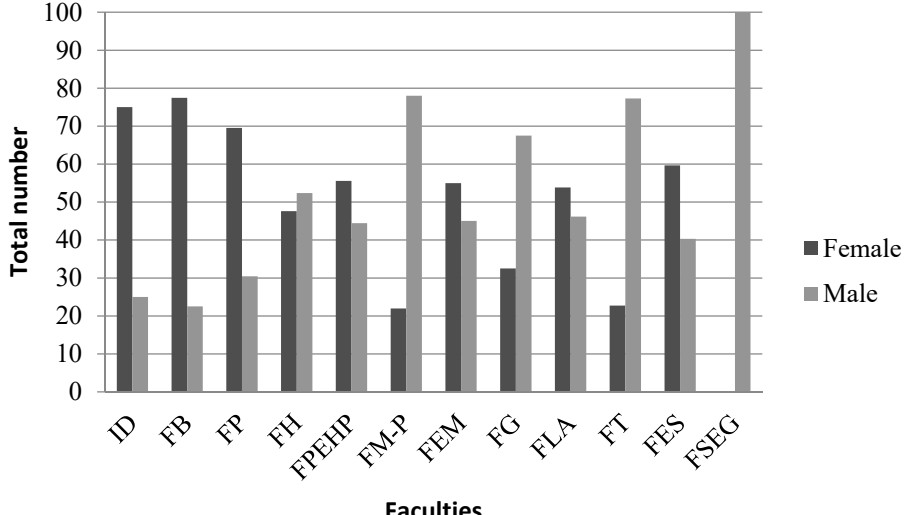

**Figure 2.** The gender structure of employment in the faculties of the University of Szczecin. Source: own study.

The employment is gender balanced only in some of the faculties, e.g., Faculty of Humanities, Faculty of Physical Education and Health Promotion, Faculty of Economics and Management, and Faculty of Law and Administration. At the same time, three units of the US are strongly feminized (Faculty of Biology, Faculty of Philology and Inter-Faculty Department), while in the other four units, mostly men are hired (Faculty of Mathematics and Physics, Faculty of Geosciences, Faculty of Theology and Socio-Economic Faculty in Gorzów Wielkopolski).

Trying to deepen the analysis of gender diversity in the US, the author conducted the research related to job positions. The structure of job positions (calculated in full-time job positions) and the percentage of women in each type of position are shown in Table 3.

**Table 3.** The employment in particular job positions in the University of Szczecin, on May 2018.

|  | Assistants | Tutors | Tutors with habilitation | Associate Professors | Full Professors |
|---|---|---|---|---|---|
| The number of posts in total | 221.5 | 689 | 71 | 344 | 121 |
| Of whom are women (%) | 61.38 | 58.4 | 44.9 | 46.1 | 31.5 |

Source: own study.

It can be observed that the structure of job positions has very little diversification, as most of the employees are hired as tutors (tutors and tutors with habilitation). This can cause the problem of aging and the lack of staff replacement in the future, as only 15.31% of employees are assistants. Considering the gender issue, it can be observed that higher job positions (tutors with habilitation, associate professors and full professors) are occupied mainly by men, and lower job positions are more

feminized. At the same time, it is worth mentioning that the observation is not always true when analyzing the structure of employment in particular faculties (Figure 3).

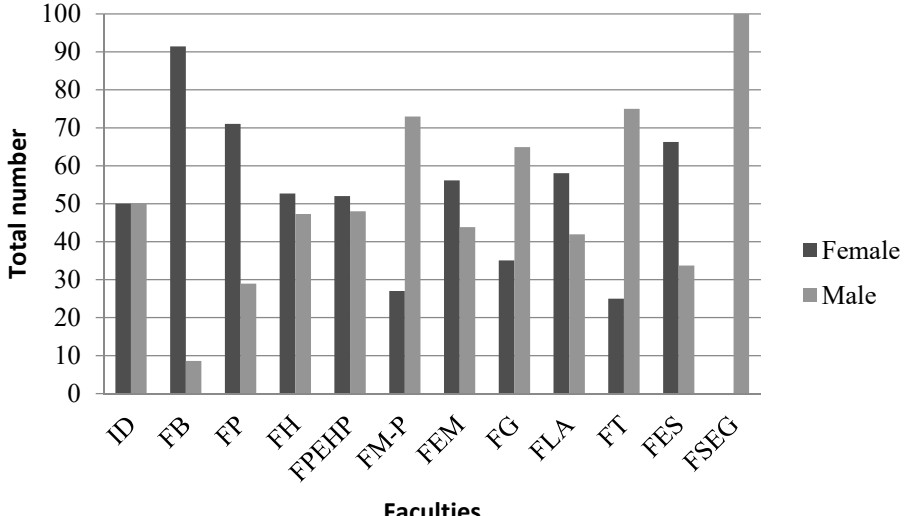

**Figure 3.** The gender structure of employment in associate professors and full professors positions in the faculties of the University of Szczecin (US). Source: own study.

Researched on collected data, it could be said that the gender structure of employment in job positions is not balanced in some faculties in the US. In the Faculties of Humanities, Theology and Philology, most of the assistants are women.

On the contrary, in the Faculty of Biology, most of the assistants are men. Tutors and tutors with habilitation are more often women in the Faculties of Biology, Philology and Economics of Services. In the Faculties of Mathematics and Physics, Theology and Socio-Economic in Gorzów Wielkopolski, mostly men work in the mentioned positions. The biggest disproportion considering gender diversity is observed in reference to the positions of professors. In seven faculties, more men than women are associate professors or full professors. Only in the Faculty of Economics and Management is gender diversity is balanced. According to the data collected in a survey, the representative of the HR department stated that the gender structure of employees in the US is a result of the demographic situation in the labor market. To secure gender-diverse work-teams, the US obeys antidiscrimination law[6]. Gender diversity is an effect of the diversity in the labor market (Table 4).

Referring to the model of organizational maturity in gender diversity management, the University of Szczecin is in the preliminary stage of GDM. This is because it presents neutral attitudes toward the diversity issue (there is no strong commitment to the idea of diversity, organizational culture is not diversely oriented and diversity is perceived as neither good nor bad) and operates in accordance with law regulations (it fulfills the necessary condition for introducing GDM). This means that the structure of employment is an effect of demographic changes both in the labor and customer markets; therefore, no specific actions, tools and programs have been introduced to manage gender diversity in the US. Consequently, the policy of GDM is based mostly on law regulations but does not have strategic meaning for managers.

---

[6]　In the academic year 2019/2020, the internal antidiscrimination regulation was introduced, called "Regulamin Pracy w Uniwesytecie Szczecińskim w Szczecinie": http://usz.edu.pl/wp-content/uploads/Zarządzenie-nr-132.2019-za\T1\l.-Regulamin-pracy-w-US-1.pdf. The new regulations were introduced on 13.09.2019, and for this reason, they are not widely discussed, as the data for article were gathered before that date.

**Table 4.** The US respondents' answers given in a survey.

| Question | Given Answer |
| --- | --- |
| Gender diversity among teachers hired in our university is . . . | a result of demographic situation in the labor market |
| To secure gender diversity within work-teams in our university, we . . . | obey antidiscrimination law |
| Gender diversity is . . . | an effect of the diversity of the labor market |

Source: own study.

### 3.2.2. Case Study 2: The University of Cordoba

Socio-Economic Conditions in Spain

Based on the data of the World Economic Forum, Spain was ranked 29 worldwide in gender equality for 2016, moved to 24 in 2017 and dropped to 29 in 2018 (Worldwide Economic Forum 2016, 2017 and 2018). More women than men live in Spain, as there are 1.04 females for each male. Less women than men of working age (15–64) participate either by working or looking for work in the labor market. Quite a considerable disproportion can be observed concerning wage equality, as the indicator is 0.5. This means that men earn more than women for the same type of work. Worryingly, women's estimated earned income (US$) is 34.03% lower in comparison to men's income, and monthly earnings (in local current) is 23.8% lower for women than for men. Men more often than women are employed in senior positions, as 69.4% of legislators, senior officials and managers are men. On the contrary, women spend about 3.3% more time than men working and doing more unpaid work. Eighty percent of the members of the board of directors of each company in the OECD ORBIS database economic are men, while only 20% of them are women. The index of the ability of women to rise to positions of leadership[7] in Spain is on the level of 0.52, which indicates the existence of a gender gap in this instance (Worldwide Economic Forum 2018).

The Characteristic of Spanish Academic Sector

Most Spanish women graduate in the education sector (22.9%), and the least in the agriculture, forestry, fisheries and veterinary sectors (0.8%). Most men graduate in engineering, manufacturing and construction (27.4%), and the less popular are agriculture, forestry, fisheries and veterinary (1.5%) among men. Almost all Spanish women who have left secondary school in the last five years (regardless of age) enroll in tertiary education (as an indicator for women is 99.2% and for men 83.5%).

According to Ministerio de Ciencia, Innovación y Universidades (2019), in 2016–2017, there were 120,383 teaching and research staff (PDI),[8] which is 1.9% more than in the previous year. In total, 102,297 PDI employees work in public universities (40.8% of them are women) and 18086 in private ones (44% of them are women). The gender structure of full-time employed PDI staff is presented in Table 5. The employment is gender balanced only in the case of non-civil-servant positions (contratados/as). The civil servant[9] positions are more or less dominated by men. In addition to this, most women are in the

---

[7]    The index is based on the response to the following survey question: In your country, to what extent do companies provide women the same opportunities as men to rise to a position of leadership? The index is on a 0–1 scale (0 is worst, and 1 is best score).

[8]    Who enjoy nearly unconditional tenure form of an early stage in academic career, and various categories of salaried employed staff (or non-civil servant staff) (Santiago et al. 2009). The career path in Spain in academic sector is as follows: (1) PhD candidate/research assistant ("becario/a de investigación" or "ayudante"); (2) postdoctoral researcher ("profesor ayudante doctor"); (3) lecturer ("contratados/as doctor"); (4) professor/a B ("profesor/a titular"/associate professor); (5) professor/an A ("catedrático"/full professor).

[9]    Civil servant academic staff: full professors—Catedrático de Universidad (CU), associate professor—professor titular (TU), Catedraticos de Escuela Universitaria (CEU), college professor—Titulares de Escuela Universitaria (TEU) (Santiago et al. 2009).

group of employees under 30 years old. The older the employee age groups are, the less women are within them (in the group of employees over 60 years old, only 28% of them are women).

**Table 5.** The gender structure of employment, considering job position in the higher-education sector, in academic year 2016/17, in Spain.

| | **Civil Servant Positions** | | | **Non-Civil-Servant Positions** | **Other** |
|---|---|---|---|---|---|
| Catedráticos/as de Univesidad CU (Full Professors **A**) | Titulares de Universidad TU (Associate Professors **B**) | Catedráticos/as de Escuela Universitaria CEU (Associate Professors) | Titulares de Escuela Universitaria TEU (College Professors) | Contratados/as CON (PhD Assistants, PhD Lecturers, Non PhD Lecturers) | Emeritos EM |
| 10,017 | 28,057 | 863 | 4284 | 52,847 | 694 |
| 21.3 | 40.3 | 31.0 | 40.2 | 44.7 | 25.8 |

Source: Ministerio de Ciencia, Innovación y Universidades (2019).

Gender Diversity Management in the University of Cordoba

The University of Córdoba (UCO) is a public High Education and Research Institution in Andalusia, in the South of Spain. It was founded in 1972 and offers undergraduate and postgraduate studies in humanities, social sciences, health sciences, natural sciences and engineering. The university is structured in three main campuses: the Humanities and Legal and Social Sciences Campus, integrated in the urban center; the Health Sciences Campus, in the west part of the city; and the Agrifood, Science and Technology campus of Rabanales, in the east part of the city. The UCO also contains the Polytechnic School of Bélmez, situated seventy kilometers away from Córdoba, where mining engineering and public works technical engineering degrees are offered. It has 10 schools and faculties: Higher Technical School of Agricultural and Forest Engineering (HTSAFE), Higher Polytechnic School of Bélmez (HPSB), Cordoba Higher Polytechnic School (CHPS), Faculty of Science (FS), Faculty of Education Sciences (FES), Faculty of Labor Science (FLS), Faculty of Philosophy and Letters (FPL), Faculty of Medicine and Nursing (FMN), Faculty of Veterinary Science (FVS) and Faculty of Law and Business and Economic Sciences (FLBES) (About UCO).

In the academic year 2018/19, 1454 researchers and teachers were hired at the University of Cordoba (38.1% women, 61.9% men), and 17,837 people studied in the University. The gender structure of employment is not gender balanced, as more men than women work in the UCO. This is also the case for most faculties, as only one faculty (Faculty of Education Science) is dominated by women (Figure 4).

The structure of job positions (calculated in full-time positions) is shown in Table 6. One conclusion is that the gender structure is not balanced in job positions in the UCO.

**Table 6.** The employment on particular job positions in the University of Cordoba, on July 2019.

| | CU | TU | CEU | TEU | CON | EM |
|---|---|---|---|---|---|---|
| Grand total | 251 | 345 | 18 | 29 | 810 | 1 |
| Of whom are women (%) | 24.70 | 41.45 | 27.78 | 41.38 | 40.99 | 0.00 |

Source: own study.

Women are in a minority in the case of all job positions. There are 34.2% of women in civil servant position in the UCO, and almost 41% in non-servant positions. The largest gender gap can be observed for the highest position, that of full professors (as only 24.7% of them are women). A similar situation occurs in almost all faculties of the UCO. Only the Faculty of Education Science is an exception, as more women (64.86%) than men (35.14%) are hired in civil servant positions (full professors, associate professors and university tenured lecturers) (Figure 5).

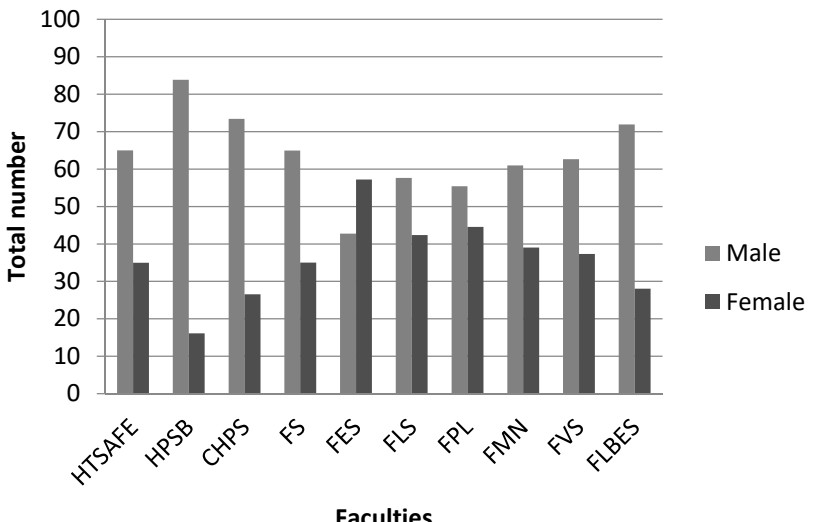

**Figure 4.** The gender structure of employment in the faculties of the University of Cordoba. Source: own study.

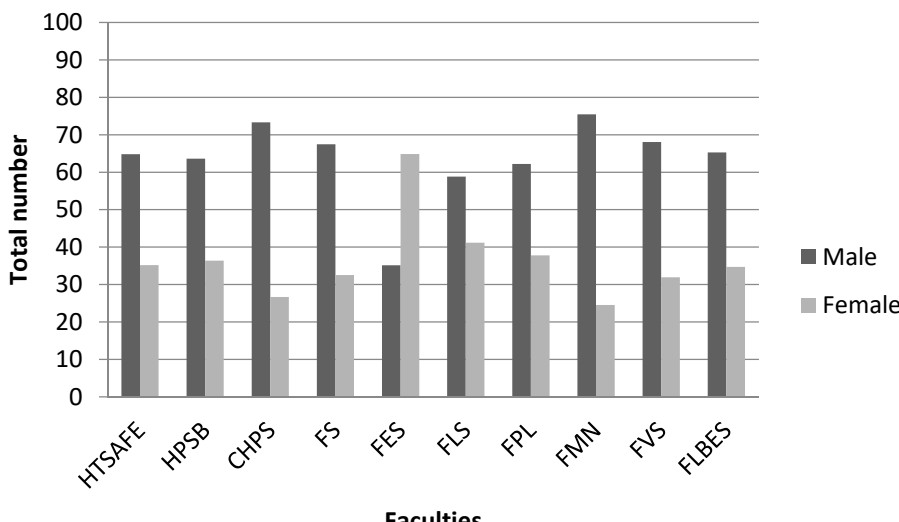

**Figure 5.** Gender structure of employment in positions of civil servant in the faculties in the University of Córdoba (UCO). Source: own study.

According to the data collected in the survey, the gender structure of employment at the UCO is a result of a demographic situation in the labor market, and gender diversity is an effect of the diversity in the labor market, so no specific action, tools and programs are introduced to manage gender diversity in the UCO. To secure gender diversity, there was internal antidiscrimination regulation created and introduced in the UCO (Table 7).

Based on the assumptions of the model of organizational maturity in gender diversity management, the University of Cordoba is classified as the one in the preliminary stage of GDM. The evidence for that is as follows: The managers created internal antidiscrimination regulations, but at the same time, diversity is not a part of organizational culture or important value. Moreover, diversity is not perceived as a source of competitive advantages but just as a consequence of the situation in the labor market.

**Table 7.** The UCO respondents' answers given in the survey.

| Question | Given Answer |
|---|---|
| Gender diversity among teachers hired in our university is … | a result of demographic situation in the labor market |
| To secure gender diversity within work-teams in our university, we … | create internal antidiscrimination regulations |
| Gender diversity is … | an effect of the diversity in the labor market |

Source: own study.

## 4. Results and Discussion

Table 8 compares the two discussed case studies. Firstly, some data on external environment are presented, and after that, information for the US and the UCO is shown.

**Table 8.** Comparison of two case studies.

| | Poland | Spain |
|---|---|---|
| Gender Gap Score | 42 | 29 |
| Sex ratio | 1.07 | 1.04 |
| Labor force participation | 62.5% | 69.3% |
| Wage gap | 0.56 | 0.50 |
| Estimated earned income | 35.16% lower than for men | 34.03% lower than for man |
| Positions of legislators, senior officials and managers | 41.2% | 30.6% |
| Minutes of unpaid work per day | 60.0 | 51.2 |
| Members of board of publicly traded companies | 20% | 20% |
| The most popular field of graduation | Business, administration and Law | Education |
| The least popular field of graduation | Information and Communication Technologies | Agriculture, Forestry, Fishery and Veterinary |
| Number of employees in academic sector | 95,400 (of which 45% are women) | 120383 (of which 40.8% are women) |
| % of women on the highest position of Full Professors | 30.0% | 21.3% |
| | **University of Szczecin** | **University of Cordoba** |
| Established in | 1984 | 1972 |
| Number of faculties | 11 | 10 |
| Number of employees | 1097 (54% are women) | 1454 (38.1% are women) |
| Gender structure of employment in the faculties | 3 faculties are feminized, 4 faculties are dominated by men, 2 faculties are gender balanced | 9 faculties are dominated by men, 1 faculty is feminized |
| Gender structure of employment considering job positions | Not balanced—31.5% of full professors are women | Not balanced—24.7% of full professors are women |
| Activities that support gender diversity | No specific activities—as gender diversity is a result of diverse labor market | No specific activities—as gender diversity is a result of diverse labor market |
| The character of GDM policy | Not strategic as the policy is a result of legislation | Close to EEO approach as internal antidiscrimination procedures are introduced |
| The attitude toward GDM | Neutral—as gender diversity is a natural effect of the labor market structure | Neutral—as gender diversity is a natural effect of the labor market structure |
| The stage of maturity in DM | Preliminary stage | Preliminary stage |

Source: own study.

Looking at socio-economic conditions in both countries, the following can be stated:

1. Polish universities function in a society with a bigger gender gap and lower women's participation in the labor market than in the case of Spain. At the same time, Polish women work more unpaid hours.

2.  On the other hand, Polish women, more often than Spanish ones, occupy higher positions, such as legislators, senior officials and managers. Additionally, the wage gap between men and women is lower in Poland than in Spain.

3.  In both cases, there are more women than men in the societies, but, at the same time, they are a minority in being members of boards of publicly traded companies.

In conclusion, both societies and economies still have barriers that cause inequalities between men and women in the labor market.

When it comes to the situation in the academic sector, it can be noticed that both Polish and Spanish women are more interested in studying humanities than technical courses, which affects their future career path. The ratio of women hired as teaching and research staff is higher for Poland. Polish women, more often than Spanish ones, are hired in the highest position of full professors.

When analyzing the two universities, it can be observed that both of them are quite new ones (as established in the second half of the 20th century), so they do not have a long history or tradition, but they play an important role in socio-economic development of the regions they operate in. They have a similar number of faculties, but the UCO is a little bit bigger, hires more employees and has more students on its campus. Summing up, it can be said that the US and the UCO are quite similar in size, thus enabling the author to compare them in terms of the gender issue. Based on the data conducted from the primary data analysis process, the author observes that employment is more gender balanced in the US than in the UCO. At the same time, the situation is diverse in the faculties of the US, while most of UCO's faculties are dominated by men. Both universities face the same problem—too little representation of women in the highest job positions of full professors. Considering the policy of GDM, both universities obey antidiscrimination law (established on the governmental level), and additionally, the UCO introduces internal antidiscrimination rules. The attitude toward GD can be described as neutral in both cases, as the diversity of work-teams is perceived as a result of demographic changes and a situation in the labor market. No specific action or strategic policy has been introduced to increase the gender diversity of employees in both universities. This discussion shows that neither the University of Szczecin, nor the University of Cordoba can be considered a diversity-oriented organization. Firstly, gender diversity is not a long-term goal on any of mentioned universities. Similarly, the organization culture is not based on the absence of discrimination. Finally, organizational objectives are not formulated to avoid the domination of a particular gender group of employees at any organizational level. For the reasons mentioned, both the US and the UCO are classified as those in the preliminary stage, according to the model of OMDM, as compliance with law is a necessary but not sufficient condition for active DM.

## 5. Conclusions

The author is aware of the limitations of the conducted research. Firstly, a comparative case study is introduced as a method of research, and for this reason, conclusions cannot be generalized for the whole population. On the other hand, the article aimed to analyze particular cases, not the whole group of universities. Secondly, critics could claim that it is difficult to compare academic sectors from different countries, as the law regulations are different and the career path is not the same for each country. That is why the author compared gender structure in job positions by grouping them in the categories of highest and lowest, to check whether there is a gender gap in the case of the highest positions in the academic sector.

The author feels further research is needed and would like to deepen the research to try to find reasons for the small representation of women in the academic sector. Additionally, it would be recommended to introduce the method of multiple case study to present a qualitative and quantitative method of data analysis.

The article makes several contributions. Theoretically, this study is important because it presents a new theoretical model of organizational maturity in diversity management (OMDM), and it contributes to a discussion on the issue of gender diversity. It is worth identifying that still there are not enough

research results on gender diversity in the academic sector. What is more, there is no article that compares the situation in Poland and Spain. For this reason, the article presented by the author can fulfill the research gap. Practically, the study describes the real situation considering gender diversity management (GDM) in two universities. Additionally, the model of OMDM can be a useful tool to assess the policy of gender diversity in different organizations.

**Funding:** This research received no external funding

**Conflicts of Interest:** The author declares no conflict of interest.

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
