# Peer review of "Gender Diversity in Academic Sector—Case Study"

_admsci, doi:10.3390/admsci10030041_

Round 1

Reviewer 1 Report

1. The topic is of importance, and  I agree that there has been too little research on gender diversity management in the academic sector.  

1.  The literature survey is very comprehensive -- in fact, too comprehensive.  It needs to be cut back to focus specifically on studies on the academic sector or directly relevant to the academic sector. 

2. Section 4 refers to gathering primary data through surveys, but I never saw any of those data used in the analysis.  Instead, the conclusions seemed to appear out of no-where, without empirical support.  The general tables giving information on the status of women in the 2 countries in general and the academic sector in the 2 countries in general is very useful and interesting context, but where is the analysis of these universities per se?

3. In particular, before you conclude that the 2 universities are at the  preliminary stage in terms of maturity, you need to explicitly show the reader the defining characteristics of each stage and line up the data to show what evidence leads you to place each university at which stage.

4. I realize how very difficult it is to write a complicated text in a language not your own, and I commend your excellent effort.  However, the manuscript needs to be edited by a native English speaker.   

Author Response

The cover letter

Author’s name: Anna Wieczorek-SzymaÅ„ska

The title of the article: Gender Diversity in Academic Sector – Case Study

Review 1.

Reviewer’s suggestions:

  1. The topic is of importance, and I agree that there has been too little research on gender diversity management in the academic sector.  
  2. The literature survey is very comprehensive -- in fact, too comprehensive.  It needs to be cut back to focus specifically on studies on the academic sector or directly relevant to the academic sector. 
  3. Section 4 refers to gathering primary data through surveys, but I never saw any of those data used in the analysis.  Instead, the conclusions seemed to appear out of no-where, without empirical support.  The general tables giving information on the status of women in the 2 countries in general and the academic sector in the 2 countries in general is very useful and interesting context, but where is the analysis of these universities per se?
  4. In particular, before you conclude that the 2 universities are at the  preliminary stage in terms of maturity, you need to explicitly show the reader the defining characteristics of each stage and line up the data to show what evidence leads you to place each university at which stage.
  5. I realize how very difficult it is to write a complicated text in a language not your own, and I commend your excellent effort.  However, the manuscript needs to be edited by a native English speaker.
  6. The text presents some writing mistakes, especially on the abstract, that need to be solved. It is necessary to improve the writing and correct both grammatical and syntactic errors. The organization of ideas could also be improved.

My answers for the Reviewer:

I would like to thank  You very much for all suggestions given to me. I did my best to meet all requirements. My answers are as follows:

Ad 1. Suggestion that do not need changes in the article

Ad 2.I have changed the part of Introduction section that refers to the literature review (lines: 175-212). I have cut back the text that described the issue of gender diversity and organizational productivity, performance, firm’s innovation, glass ceiling phenomenon. I have added 4 literature position to the section of literature review. The additional  literature is on gender diversity in academic sector. These are:

  • 2017. Cracking the code: Girls' and womens' education in Science, Technology, Engineering and Mathematics (STEM), Paris: United Nations Educational, Scientific and Cultural Organization, 85 p.
  • Howe-Walsh, Liza and Turnbull, Sarah. 2016.Barriers to women leaders in academia: tales from science and technologies. Studies in Higher Education 41 (3): 415-28.
  • Directorate-General for Research and Innovation (European Commission). 2019. She Figures 2018. Gender in Research and Innovation. Statistics and Indicators, Luxemburg: Office of the European Commision, 216 p.
  • Claeys-Kulik, Anna L., Ekman Jørgensen, Thomas, and Stöber, Henriette. 2019. Diversity, Equity, and Inclusion in European Higher Education Institutions. Results from the INVITED Project. Brussel: European University Association Asil, 51p.

Ad 3. In the section Methods and Materials (previously section 4, after changing the order section 2). I have added the description of questions asked in the survey (lines: 238-253). In the section on results (previously section 2 after changes section 3 called “Analyzing Organizational Maturity in Gender diversity Management”). I have added that the information about respondents (lines: 254-255).

The general information about the gender structure of employment are placed in table: 2and 5 but Tables: 3,4, 6, 7 and Figures: 2,3,4,5 present analysis of each university. Information for Tables: 3,4,6,7 and Figures: 2,3,4,5 was gathered through survey. Tables: 3, 6  and Figures: 2,3,4,5 present gender structure of employment on universities. The idea was to check whether the structure of employment is gender balanced or not. Tables: 4 and 7 were added to present answers retrieved from respondents in a survey (lines: 503 and 592). The idea was to characterize the policy and organizational attitude toward gender diversity management. In consequence a part of the table 8 was created and conclusions were made. So that there is an analysis of particular universities per se.

Ad 4. I have specified definitions of preliminary and mature gender diversity management in the section 3.1 (previously 2.1) lines 381- 396. Additionally I have defined the condition to move organization from preliminary stage of GDM toward mature GDM (lines 397-403). Moreover I have added Tables 4 and 7 to quote respondents’ answers in a survey (lines: 503 and 592) and I have interpreted those answers considering the assumptions of the model of organizational maturity in gender diversity management (lines: 505-513 and 594-599). This was the basis to classify universities as institutions on a preliminary stage of GDM.

Ad 5. The text was edited by the native speaker after Your comments. The editor was Mr. Mark Fitzpatrick (Ireland). Lines with edited text: 21-3, 53, 55, 58, 59, 62, 66, 81, 82, 84, 87, 88, 90, 91, 93, 94, 96, 98, 99, 108, 112, 116, 214-217, 259-265, 282, 283, 292, 298, 311, 313, 318, 319, 401-403, 444, 445, 474, 479, 485-490, 518, 521, 529, 542, 545, 606, 607, 614, 617-619, 622-626, 674, 680, 682.

Ad 6. I have rewritten the abstract and formulated my ideas in a different way (lines: 21-43).

Reviewer 2 Report

The work is very interesting and presents some remarkable conclusions. However, some aspects need to be improved.

The text presents some writing mistakes, especially on the abstract, that need to be solved. It is necessary to improve the writing and correct both grammatical and syntactic errors. The organization of ideas could also be improved.

Line 7: characteristic should by characteristics

Line 8: I would use a “,” after “At the same time”

Lines 11-13: The sentences “To evaluate organizational maturity in gender diversity management. In the empirical part the Author uses the case study method” have no sense, may be the “.” should be a “,”.

Lines 25-27: “A nature determines individual’s sex and by this classifies a person as a man or a women (from the biological point of view), it also defines a gender (the social point of view) by defying particular social roles that are typical for a man or a woman in a society.” Does the author mean that nature defines gender? There is also a typo in the term “defying”, I supposed it would be “defining”.

Line 29: change “In many cases is also leads” to “In many cases it also leads”

Line 48: is “(as of October-December 2019)” a cite? I cannot find the reference. Perhaps it is the time in which the search was carried out. In that case, the phrase should be rewritten.

Lines 50-51: please, include references to these methods.

About sections names and order: I suggest changing the name of section 2 (Results) to something like “Analyzing Organizational Maturity in Gender Diversity Management” or “Case Study”. Moreover, the order of the sections needs to be review. Explanation in lines 483 to 489 is crucial for paper understanding and should have been previously introduced, just before starting to talk about these universities. In addition, there are two sections 2 and no section 3 and two sections 4 and no section 5.

I suggest the following section order for a better understanding.

Section 1: Introduction

Section 2: Materials and methods.

Section 3: Analyzing Organizational Maturity in Gender Diversity Management

Section 4: Results and Discussion

Section 5: Conclusions

Lines 233-235 format should be change form center to justified.

Introduction: I would recommend adding the following references

UNESCO 2017, Cracking the code: Girls' and womens' education in Science, Technology, Engineering and Mathematics (STEM)

  1. Howe-Walsh and S. Turnbull 2016, Barriers to women leaders in academia: tales from science and technologies

Directorate-General for Research and Innovation (European Commission) 2019, She Figures 2018. Gender in Research and Innovation. Statistics and Indicators

Table 1: Consider not splitting the table into two sheets. I suggest increasing width.

Lines 256-257: Incorrect use of “The more”. The correlative comparative is a paired construction where each part is syntactically alike. The more…, the … . The sentence could be rewritten as “The more proactive attitude of organizational managers towards diversity, and the more stronger strategic importance of workforce diverse, the closer to mature-oriented DM will be the organization”

Lines 259-261: Please review the sentence for a better understanding

Section 2.2. Research Findings: this section contains a lot of important data. However, I could not find any supporting reference. It is necessary to clarify data fonts and to include the corresponding references.

Figure 2: this figure includes a faculty identifier (ID) that has not been previously defined. The faculties ID and FSEG have no value represented.

Line 355: an space is missing in ”to29”

Line 356: the opposite of female is not men. Consider changing the sentence “there are 1,04 females for each man” for “there are 1,04 females for each male”

Line 379 and Table 4: Spanish language presents differentiation by gender in nouns and adjectives. In order to use inclusive language, consider changing the terms “Contratados” to “Contratados/as”, Catedraticos (which in fact is Catedráticos) to “Catedráticos/as”, Becario to Becario/a and Profesor to Profesor/a

Table 6: Consider not splitting the table into two sheets. I suggest increasing width.

Line 438: I would recommend change “Conclusion is that” to “The conclusion reached is that”

Lines 441-443: Please review the sentence for a better understanding

Line 470: “double- case” should be “double-case”

Line 482: I would change “process of organization of work” to “work organization process”

In general, I think there is an excessive use of the form “The Author…”

Nevertheless, I would like to congratulate the author for the performed study.

Author Response

Review 2.

Reviewer’s suggestions and my answers.

I would like to thank  You very much for all suggestions given to me. I did my best to meet all requirements. My answers are presented in the brackets.

Line 7: characteristic should by characteristics (done – line 21)

Line 8: I would use a “,” after “At the same time” (done – line 22)

Lines 11-13: The sentences “To evaluate organizational maturity in gender diversity management. In the empirical part the Author uses the case study method” have no sense, may be the “.” should be a “,”. (I have rewritten the abstract and formulated my ideas in a different way - lines: 21-43 .)

Lines 25-27: “A nature determines individual’s sex and by this classifies a person as a man or a women (from the biological point of view), it also defines a gender (the social point of view) by defying particular social roles that are typical for a man or a woman in a society.” Does the author mean that nature defines gender? There is also a typo in the term “defying”, I supposed it would be “defining”. (I have formulated the idea as follows: The issue of gender can be considered in a biological context (a nature determines individual’s sex and by this classifies a person as a man or a women) and in social context (focuses mainly on defining particular social roles that are typical for a man or a woman in a society – lines: 50-53).

Line 29: change “In many cases is also leads” to “In many cases it also leads” (done – line 55)

Line 48: is “(as of October-December 2019)” a cite? I cannot find the reference. Perhaps it is the time in which the search was carried out. In that case, the phrase should be rewritten. (the Author carried out the research in a period October-December 2019 line 75).

Lines 50-51: please, include references to these methods. (it should be “comparative case study”, the reference was added: Goodrick, Delwyn. 2014. Comparative Case Studies: Methodolical Briefs – Impact Evaluation No 9. Methotolical Briefs 9: 1-17 – lines 77-78)

About sections names and order: I suggest changing the name of section 2 (Results) to something like “Analyzing Organizational Maturity in Gender Diversity Management” or “Case Study”. Moreover, the order of the sections needs to be review. Explanation in lines 483 to 489 is crucial for paper understanding and should have been previously introduced, just before starting to talk about these universities. In addition, there are two sections 2 and no section 3 and two sections 4 and no section 5.

I suggest the following section order for a better understanding.

Section 1: Introduction

Section 2: Materials and methods.

Section 3: Analyzing Organizational Maturity in Gender Diversity Management

Section 4: Results and Discussion

Section 5: Conclusions

(The order and title of sections has been changed as the Reviewer suggests. The previous order was due to recommendations for Authors put on the website of the Journal).

Lines 233-235 format should be change form center to justified. (done lines 348-356)

Introduction: I would recommend adding the following references (done, moreover the introduction was changed to focus more on research from academic sector lines: 175-212).

UNESCO 2017, Cracking the code: Girls' and womens' education in Science, Technology, Engineering and Mathematics (STEM)

Howe-Walsh and S. Turnbull 2016, Barriers to women leaders in academia: tales from science and technologies

Directorate-General for Research and Innovation (European Commission) 2019, She Figures 2018. Gender in Research and Innovation. Statistics and Indicators

Table 1: Consider not splitting the table into two sheets. I suggest increasing width. (done – line 342)

Lines 256-257: Incorrect use of “The more”. The correlative comparative is a paired construction where each part is syntactically alike. The more…, the … . The sentence could be rewritten as “The more proactive attitude of organizational managers towards diversity, and the more stronger strategic importance of workforce diverse, the closer to mature-oriented DM will be the organization” (done – lines 398-401).

Lines 259-261: Please review the sentence for a better understanding (I have changed the piece of text to clarify my way of thinking. . I have specified definitions of preliminary and mature gender diversity management in the section 3.1 (previously 2.1) lines 381- 396. Additionally I have defined the condition to move organization from preliminary stage of GDM toward mature GDM (lines 397-403).

Section 2.2. Research Findings: this section contains a lot of important data. However, I could not find any supporting reference. It is necessary to clarify data fonts and to include the corresponding references. (All tables and Figures have sources. There are references in text).

Figure 2: this figure includes a faculty identifier (ID) that has not been previously defined. The faculties ID and FSEG have no value represented. (I have changed the figure. All data is now correct, I have also explained what ID means – lines: 451 and 459).

Line 355: an space is missing in ”to29” (done – line 517).

Line 356: the opposite of female is not men. Consider changing the sentence “there are 1,04 females for each man” for “there are 1,04 females for each male” (done – line518)

Line 379 and Table 4: Spanish language presents differentiation by gender in nouns and adjectives. In order to use inclusive language, consider changing the terms “Contratados” to “Contratados/as”, Catedraticos (which in fact is Catedráticos) to “Catedráticos/as”, Becario to Becario/a and Profesor to Profesor/a (done – line 549)

Table 6: Consider not splitting the table into two sheets. I suggest increasing width. (now it is table 8. I have changed the width – line 603).

Line 438: I would recommend change “Conclusion is that” to “The conclusion reached is that” (done- line614)

Lines 441-443: Please review the sentence for a better understanding (done – lines 618-619)

Line 470: “double- case” should be “double-case” (it was changed into comparative case study – line667)

Line 482: I would change “process of organization of work” to “work organization process” (done – line 258).

In general, I think there is an excessive use of the form “The Author…”

Nevertheless, I would like to congratulate the author for the performed study.

Reviewer 3 Report

Minor spell check required!

Author Response

The cover letter

Author’s name: Anna Wieczorek-SzymaÅ„ska

The title of the article: Gender Diversity in Academic Sector – Case Study

Reviewer’s suggestions and my answers.

I would like to thank  You very much for all suggestions given to me. I did my best to meet all requirements.

1.Reviewer’s suggestion: Minor spell check required!

My response:

The text was edited by the native speaker (Mr. Mark Fitzpatrick from Ireland). Lines with edited text: 21-3, 53, 55, 58, 59, 62, 66, 81, 82, 84, 87, 88, 90, 91, 93, 94, 96, 98, 99, 108, 112, 116, 214-217, 259-265, 282, 283, 292, 298, 311, 313, 318, 319, 401-403, 444, 445, 474, 479, 485-490, 518, 521, 529, 542, 545, 606, 607, 614, 617-619, 622-626, 674, 680, 682.

Round 2

Reviewer 1 Report

I appreciate the significant amount of effort the author(s) have invested in grappling serious with reviewer comments.  I belive the manuscript is considerably stronger as a result. I particualrly note the more extensive discussion to the survey and to the more explicit description of the stages of maturity. 

Unfortunately, the article is still very far from publishable quality. 

First, the writing is still terribly wordy, often awkward, and "not quite English." 

More importantly, while the survey has been more explicitly described, the results are inadequately presented.  For example, when Table 7 reports survey respondents' answers to the question about why the university has undertaken gender diversity efforts, all that is reported is what apprears to be the most common answer.  Normal scholarly practice would be to report the %s for all answers. 

In a similar vein, I am completely mystified about where the survey data referring to US reposnes comes from (Table 4).  

Furthermore, if this is to be a comparative case study, the survey results from the two universities would need to be reported side by side in the same table, so that they can be compared.   (This comment applies not only to tables but also text.  Right now, the paper reads largely as two separate case studies, with little conmparison between them. 

Next, the discussion of conceptual models of diversity(Section 3.1) is weak.  For example, I was partucualry upset by the following statement: "Since the 60's there has been a strong belief in White Supremacy so that immigrants and members of ethnic mionorities face prejucice."  This statement is so wrong in so many ways that I hardly know where to begin, but I will mention in particular (a) this problem hardly began in the 60's but has been an issue for all several undred years of American society, and (b) the phrase White Supremacy has an entirely different meaning than the one implied here.  The same messiness applies to the folowing sentence, which starts to discuss women, and throughout the whole literature survey.

I could go on about nearly every paragraph in the paper.  

Author Response

Anna Wieczorek-Szymańska

Gender Diversity in Academic Sector – case study.

Again I would like to thank to the Reviewer for all comments and suggestions. I hope this time I met all the requirements.

  1. Reviewer: I appreciate the significant amount of effort the author(s) have invested in grappling serious with reviewer comments.  I belive the manuscript is considerably stronger as a result. I particualrly note the more extensive discussion to the survey and to the more explicit description of the stages of maturity. Unfortunately, the article is still very far from publishable quality. 

My response: none

  1. Reviewer: First, the writing is still terribly wordy, often awkward, and "not quite English." 

My response: the article was checked and edited by two native speakers one from Ireland one from England.

  1. Reviewer: More importantly, while the survey has been more explicitly described, the results are inadequately presented.  For example, when Table 7 reports survey respondents' answers to the question about why the university has undertaken gender diversity efforts, all that is reported is what apprears to be the most common answer.  Normal scholarly practice would be to report the %s for all answers. 

My response: both tables 4 and 7 present responses of respondents from both universities. I cannot present % of responses as in case of both universities only one respondent fulfilled the survey. The goal was not to check employees’ opinion on gender diversity management but to get information on real activities undertaken on the universities. For this reason I asked top managers about organizational attitudes and policy in GDM. In case of University of Szczecin the respondent was vice-Rector, in case of university of Cordoba it was the head of HR department. (additional information about respondents  - lines: 185-188).

  1. Reviewer: In a similar vein, I am completely mystified about where the survey data referring to US reposnes comes from (Table 4).  

My response: I explained it in the paragraph 3.

  1. Reviewer: Furthermore, if this is to be a comparative case study, the survey results from the two universities would need to be reported side by side in the same table, so that they can be compared.   (This comment applies not only to tables but also text.  Right now, the paper reads largely as two separate case studies, with little conmparison between them. 

My response: The case study in not only about the universities but also about the whole academic sector and the socio-economic environment in the both countries. It would e difficult to present both case studies in one place so that the table 8 presents the results in a comparative way. I added three lines into the table 8 to compare the statements of the managers on both universities. I also added some more conclusions – lines: 549-556

The character of GDM policy

Not strategic as the policy is a result of legislation

Close to EEO approach as internal antidiscrimination procedures are introduced

The attitude toward GDM

Neutral – as gender diversity is a natural effect of the labor market structure

Neutral – as gender diversity is a natural effect of the labor market structure

The stage of maturity in DM

Preliminary stage

Preliminary stage

  1. Reviewer: Next, the discussion of conceptual models of diversity(Section 3.1) is weak.  For example, I was partucualry upset by the following statement: "Since the 60's there has been a strong belief in White Supremacy so that immigrants and members of ethnic mionorities face prejucice."  This statement is so wrong in so many ways that I hardly know where to begin, but I will mention in particular (a) this problem hardly began in the 60's but has been an issue for all several undred years of American society, and (b) the phrase White Supremacy has an entirely different meaning than the one implied here.  The same messiness applies to the folowing sentence, which starts to discuss women, and throughout the whole literature survey.

My response: I edited paragraph 3.1:

  • I have changed the sentence about white supremacy line:201-203 (The concept of Diversity Management has its roots in Equal Employment Opportunity (EEO) and Affirmative Action (AA) approaches. In turn, these concepts were the outcomes of Civil Rights Movement that took place in the USA in the 20th century (Shore et a. 2009)).
  • I have added the information on Report ‘Workforce 2000’ – lines: 214-217 (In 1987 the report ‘Workforce 2000’ by Hudson Institute presented demographic shifts according to which white male would no longer be a majority in the workforce due to the total increase of women and other minorities in the labor market (Johnston and Packer 1987)).
  • I have deleted the section about dissimilarities between employees as it was too long - (deleted text: In turn Fazlagić (2014) discusses people’s differences in terms of so called primary, secondary and organizational factors. Primary criteria like the race, ethnicity, age, sexual orientation are beyond the control of the individual while secondary criteria e.g.: the level of education, place of living, family status, language, religion et al. can be changed by individuals more or less consciously during lifetime. Organizational variables can segregate employees from each other on the bias of.: seniority, job position, sector and types of employment et al. Loden and Rosener (1991) present the idea of so called diversity wheel to show four dimensions of differences and similarities between people in organization. These are: Personality (the temperament, the character and attitudes of individuals), Internal dimension (age, sex, ethnicity, race), External dimension (education, family status, the level of income), Organizational dimension (hierarchical and functional aspects of work like job position, range of power in organization, performed function). To sum this up:  organization’s employees differ from each other in many ways.)
  • I have changed the place of the table 1 – now it describes approaches to DM in general not just do GDM. Line 253. And I have corrected the description of the models presented in the table 1.
  • I have added more description on mature GDM – lines:300-308.
  • I have added two more references:

Kandola, Rajvinder, and Fullerton, Johana. 1998. Diversity in action: managing the mosaic. 2nd ed. London: Chartered Institute of Personnel and Development (CIPD), 192 p.

Johnston, William B. and Packer, Arnold H. 1987. Workforce 2000: Work and Workers for the 21st Century. Indianapolis: Hudson Institute, 143 p.
